# Health Professionals’ Knowledge and Views on the Use of Convenience Cooking Products: An Australian Cross-Sectional Study

**DOI:** 10.3390/nu17071156

**Published:** 2025-03-26

**Authors:** Natasha Brasington, Emma L. Beckett, Taiwo O. Akanbi, Penta Pristijono

**Affiliations:** School of Environmental and Life Sciences, The University of Newcastle, Ourimbah, NSW 2258, Australia; taiwo.akanbi@newcastle.edu.au (T.O.A.); penta.pristijono@newcastle.edu.au (P.P.)

**Keywords:** health professionals, convenience cooking products, food choice, cross-sectional study

## Abstract

Background/Objectives: Convenience cooking foods have gained popularity as they reduce the time and effort spent on preparation. These types of products are often deemed unhealthy and low in nutrients. However, if these products had an adequate serving and variety of vegetables and healthful sources of protein, they could be a good time-efficient and cost-effective alternative. However, there is no established evidence on health professionals’ opinions and ideas about convenience cooking products, nor is their information whether if they consume them or recommend them to their patients/clients, or whether they could provide a solution when patients are confronted with decision fatigue. The objective of the present study is to define the opinions that health professionals have regarding convenience cooking products and their healthfulness, if they use these products themselves, if they recommend these products to clients, and their ideas regarding decision fatigue and the use of convenience cooking products. Methods: A cross-sectional survey analysed the opinions of 143 Australian health professional participants, including dietitians, nutritionists, and doctors on their use of the products, if they recommend them to their clients, their health opinions of the products and decision fatigue. Results: The findings indicate that 74.8% of the participants use convenience products. The participant’s professions (*p* = 0.0014) and their personal usage of the products (*p* = 0.0154) significantly correlated with their recommending these products. Additionally, 86.7% of participants believed that decision fatigue impacts food choices. Conclusion: These insights highlight the complex role of convenience cooking products in dietary practices, particularly for time-poor individuals, and provide insight into the potential for future improvements in the nutritional formulations of these products to enhance their acceptability among health professionals.

## 1. Introduction

Convenience foods are fully or partially prepared products that require minimal effort to use and are designed to minimise the time and cognitive demands of preparing food [1]. Convenience foods are typically processed and have long shelf lives [2], and are selected based on factors such as ease of use, packaging, nutritional value, safety, variety, and aesthetic appeal [3].

Their rising popularity is linked to lifestyle changes, particularly increased workforce participation among women, leading to greater reliance on convenience foods [4]. Studies show these foods save time and reduce meal preparation effort [5]. A qualitative study involving 787 respondents examined the perceived versus actual constraints on meal preparation. The study found that a preference for convenience food is negatively related with enjoyment of cooking, involvement with food, household size, and having children, but positively associated with overall role overload. Furthermore, it was identified that individuals who prepare meals and work more than thirty hours per week are more likely to use convenience food products compared to those who work fewer than nine hours per week [6].

Another factor contributing to the increased demand is the ageing population, which has driven a preference for food products that require less cognitive effort and come in smaller, single-serving packages, thus reducing food waste [7]. A cross-sectional survey conducted in Melbourne, Australia, with 352 respondents of varying ages, found that younger adults are three times more likely to prepare convenience foods than older adults [8]. The study also reported that younger participants are over five times more likely to consume takeaway foods compared to those aged 60 years and over, with these findings that younger adults are more likely to consume convenience foods showing consistency with other, similar studies [9,10]. Additionally, small qualitative studies suggest that elderly individuals who live alone or are widowed are more likely to rely on convenience foods [8,11,12,13].

Despite their convenience, these foods are criticised for their lower nutritional value and sustainability concerns [14,15]. Ready-to-eat meals are linked to a higher fat intake and reduced fruit and vegetable consumption [16]. However, this is where convenience cooking products could be altered in order to become a healthier and more balanced option. Convenience cooking products include meal bases, recipe bases, simmer sauces, marinades, ready-made sauces, and recipe concentrates. These products come in a liquid or powdered form and contain suggested ingredients and cooking directions on the package.

Food choice behaviours are influenced by a multitude of factors, including individual characteristics, social environments, physical environments, and macro-level influences [17]. These factors can interact in complex ways, both directly and indirectly, to shape eating behaviours and food choices. Wansink et al., 2007, suggest that individuals make an average of 221 food-related decisions each day [18], encompassing not just what to eat, but also when, how, in what combination, and in which settings [19]. This highlights the significant cognitive load associated with food choices and underscores why knowledge and education alone are insufficient to improve eating habits across a population. When accounting for the impact of environmental influences and other stressors on food choices, it becomes evident that many decisions are made in distracting or stressful contexts, which can lead to poor food choices or “mindless eating” behaviours [20,21]. This could be a contributor to the current health status of Australians.

Poor diet was identified as the fifth leading global mortality risk in 2017 [22], with rising healthcare costs linked to poor nutrition [23,24]. Despite the benefits of vegetable-rich diets, 95% of Australians fail to meet the recommended daily intake [25]. When comparing trends from ABS surveys of individuals not meeting the daily recommendations for vegetable consumption from 2007 to 2022, there has been minimal change in how many people meet the dietary guidelines [26]. New strategies are needed to increase vegetable consumption. If properly formulated, convenience cooking products could help improve diet quality, particularly for time-poor individuals or those with limited cooking skills.

Healthcare professionals, including dietitians, nutritionists, nurses, and doctors, play a crucial role in providing nutrition care and advice to Australians. These professionals are generally regarded as trustworthy and credible sources of nutrition and health information, and their guidance can significantly influence patients’ dietary behaviours [27]. However, no published evidence exists on their views regarding convenience cooking products, decision fatigue, and dietary choices. This study aims to assess their opinions, their usage patterns, and whether they recommend these products to clients. A secondary objective is to explore their understanding of decision fatigue and whether these products could serve as an alternative in such cases.

## 2. Materials and Methods

### 2.1. Study Design, Setting, and Recruitment

This cross-sectional study was granted ethical approval by The Human Research Ethics Committee of the University of Newcastle on the 8 May 2024 (Reference No. H-2024-00079). Given the exploratory nature of this study and the absence of prior research in this area, a snowball recruitment method was used to recruit a convenience sample. This method was chosen due to financial constraints and the challenges associated with recruiting participants for a novel topic. The survey was designed and administered online through QuestionPro; the survey was self-administered and anonymous. The survey was advertised online for ~20 weeks (14 May 2024–10 October 2024) on the investigators’ social media pages: X (formerly known as Twitter), Instagram, and LinkedIn. Before proceeding with filling out the survey, all participants were required to sign informed consent. Participants were included in the study if they were living in Australia, over 18 years of age, and were professionals working in dietetics, nutrition, medicine, nursing, or any other allied health field where they are involved in nutrition. Any incomplete or missing values were automatically considered invalid and were not included in the study. The study followed the Strengthening the Reporting of Observational Studies in Epidemiology (STROBE) reporting guideline for cross-sectional studies [28].

### 2.2. Profession and Qualifications

Participants were asked about their profession, defining what area of health profession they were from Dietitian (accredited practicing), Dietitian (other), Nutritionist (Associate or registered), Nutritionist (other), Nurse, Medical doctor, other nutrition or allied health professional, if they were not from the following list they were excluded ending the survey. They were also asked to report the highest level of qualification relevant to their profession and their years of work experience, which was treated as a quantitative measure. 

### 2.3. Use of Convenience Cooking Products

Participants were asked about their usage of the products and how often they use them, with the following response options: “multiple times a week”, “once a week”, “once a month” or “less than once a month”. Participants were also asked if they use the product when “when I don’t feel like planning a meal”, “just because”, “on a weekly basis”, “when I’m having a busy day”, and “when I want to explore a new variety of food”.

### 2.4. Recommending These Products to Patients/Clients

Participants were asked if they currently recommend, or if they would recommend, these products to their clients or the general public. A 5-point Likert scale was used that ranged from 1 (strongly disagree) to 5 (strongly agree). The scale was used to identify their views and opinions regarding these products, with some answers being, “they contain too much sodium”, “they don’t have enough protein”, “they are not balanced”, and “doesn’t promote healthy heating”. Reverse scale questions were integrated as an attention check. The complete scales are included in the Appendix A. Cronbach’s Alpha was calculated to assess the internal validity of the scales.

### 2.5. Health Perceptions of These Products

These questions were used to gain information on health professionals’ perceptions of how these products are. Participants were asked to choose the category that best describes the overall healthfulness of convenience cooking products. A 5-point Likert scale was used that ranged from 1 (strongly disagree) to 5 (strongly agree). It featured statements such as “they can increase cooking at home”, “they can increase cooking skills”, and “they do not improve diet quality” The complete scales are included in the Appendix A.

### 2.6. Decision Fatigue

Participants were asked if they had heard of the term decision fatigue, their perceptions of different statements that may be related to decision fatigue, and if they think convenience cooking products could be a good alternative when faced with decision fatigue.

### 2.7. Statistical Analysis

Data were analysed using JMP (Pro 14; SAS Institute Inc., Cary, NC, USA). Both categorical and continuous data were used. The statistically significant threshold was a *p*-value of <0.05. Contingency tables (Pearson X2) and nominal logistical regression were used to assess the differences in distributions between categories. Standard least squares regression was used to compare adjusted least squares means by category (adjusted for age, sex, income, education, and working hours).

## 3. Results

### 3.1. Demographics

A total of 272 participants responded to the survey. One hundred and thirty-eight participants were excluded due to incomplete responses, twelve were excluded for completing the survey in less than half the median completion time (<450 s), and a further eight were excluded for not residing in Australia. Overall, there were a total of 143 participants in the final sample. Most of the participants consisted of females (94.4%) and respondents’ ages ranged from 18 to 64 years, with most participants being aged 25–34 years (54.1%) (Table 1). A master’s degree (39.2%) and a bachelor’s degree (38.4%) were the most common qualifications (Table 1). The largest group of professionals represented was dieticians (accredited practicing dieticians and others), comprising 54.8% of the total, and 94.4% of participants were actively engaging in their profession. Most professionals had 1–6 years of experience in their profession (54.5%) (Table 1).

### 3.2. Health Professionals’ Current Use of Convenience Cooking Products

A total of 107 (74.8%) of health professional participants reported using convenience cooking products. The most common frequency of the usage of convenience cooking products was once a week (39.9%), while 20.3% reported never using them (Table 2). The most common reason for using convenience products was during busy days (60.1%), followed by not feeling like planning a meal (36.4%) (Table 2).

Professional and personal use of convenience cooking products significantly predicted health professionals recommending these products to clients. Both profession (*p* = 0.0014) and product usage (*p* = 0.0154) contributed significantly to the unadjusted estimates. While dietitians were less likely to recommend these products (*p* = 0.0810) in comparison to nutritionists (*p* = 0.0750), this association was not statistically significant. Those who reported using the products were more likely to recommend them (*p* = 0.0198). A total of 64.1% of users of convenience cooking products said they recommend these products, in comparison to 41.4% of non-users who stated they recommend the products (*p* =< 0.05) (χ^2^ = 44.48, *p* < 0.0001) (Table 3).

Users of convenience cooking products generally recommend the products to their patients at least once a week (26.7%), while non-users generally recommend them less frequently, with most never recommending them (51%). When referring to the target audience for these products, both users and non-users primarily suggest these products to time-poor people and those with limited cooking skills (Table 3).

Health professionals’ perceptions of convenience cooking products differed significantly between users and non-users across several statements. Users of these products were more likely to agree that they provide adequate protein (mean = 2.6; 95% CI: 2.4–2.8) compared to non-users (mean = 2.2; 95% CI: 1.8–2.6), with the results showing marginal significance (*p* = 0.05). Non-users were more likely to agree that the products lacked sufficient fibre (mean = 4.1; 95% CI: 3.8–4.5) compared to users (mean = 3.6; 95% CI: 3.5–3.8; *p* = 0.01) (Table 4). Additionally, users were more likely to believe that convenience products can contribute to a balanced diet (mean = 2.6; 95% CI: 2.4–2.7) compared to non-users (mean = 3.1; 95% CI: 2.8–3.5; *p* = 0.0064) and that these products could be useful for weight control (users: mean = 2.8 and 95% CI: 2.6–3.0; non-users: mean = 3.3 and 95% CI: 2.9–3.6; *p* = 0.02) (Table 4).

When referring to statements about decision fatigue, 86.7% of participants believed that decision fatigue was related to poorer food choices (“Do you think that a person who suffers from decision fatigue may be more prone to poorer food choices?”). A total of 61.5% of participants agreed that convenience cooking products could be a healthier, more balanced option if a person suffers from decision fatigue, and 53.8% of participants responded yes to the statement “If you had a client/patient who was suffering from decision fatigue, would you suggest these products to them?” (Table 5).

## 4. Discussion

The present study is the first to provide insight into health professionals’ perceptions and personal use of convenience cooking products. The objectives of this study were to determine what health professionals think of convenience cooking products, if they recommend convenience cooking products to their clients or would consider recommending them, and who they think these products are suitable for. The use of convenience cooking products was common among health professionals, and they were used mostly when they had busy days. Personal use of the products among health professionals significantly contributed to their likelihood of recommending such products to patients/clients. Furthermore, 86.7% of participants believed decision fatigue impacts food choices, with over half endorsing convenience products as a balanced option for those affected by decision fatigue.

Among health professionals, 74.8% (*n* = 107) reported personal use of convenience cooking products, primarily due to busy schedules (60.1%). This is supported by previous studies linking the use of convenience foods to time-poor individuals. A study focusing on Irish consumers by Reed et al. (2000) identified that the reasons for purchasing ready meals include time pressure from work, their convenience when eating at work, or plans to use them as a backup option [29]. Studies highlight a decline in the time spent cooking. For example, a cross-sectional study by Wolfson et al. (2024) performed between 2007 and 2010 found that cooking dinner more frequently and spending more time preparing it were both associated with a lower intake of ultra-processed foods and a higher intake of minimally processed foods [30]. Overall, convenience products may help health professionals manage demanding work schedules.

The present study results also indicated a significant relationship between personal use of these products and the likelihood of recommending them to clients, with health professional users of these products being more inclined to recommend convenience cooking products, especially to people with limited cooking skills and time constraints. This raises concerns about potential bias in dietary recommendations. Users of these products may perceive them more favourably, possibly overlooking issues like high sodium or low fibre, while non-users may be overly critical. Such biases could impact patient outcomes, either by promoting suboptimal diets or dismissing practical options. Future research should investigate whether training on evidence-based assessments of convenience cooking products can help health professionals provide more balanced, individualised recommendations, minimising bias and optimising patient dietary outcomes.

Dietitians were less likely to recommend these products compared to nutritionists, although the difference was not significant; this may reflect differences in professional training and focus. Further analyses may help to define outcomes. Dietitians, traditionally focused on promoting whole, minimally processed foods, may have reservations about the nutritional quality of convenience cooking products, particularly concerns around their sodium content, protein sufficiency, and overall balance. A cross-sectional observational study by Thurecht et al. (2018) stated that dietitians were influenced by the healthfulness of packaged foods, as derived from their sugar content, total fats, sodium and saturated fat values, and the ingredients list [31]. However, food choice is never uniform and there are other environmental factors that should be considered when suggesting foods to clients/patients. This is supported by the regression analysis, which identified profession and personal use as significant predictors for recommending these products. The dietitian’s role in assisting the public with healthier food choices has become increasingly important due to the change in household roles, with more household cooks working longer hours, fewer cooking skills being taught throughout the life stages, and a growing diversity of packaged and convenience foods being available, as well as the increase in their marketing [32,33,34]. These findings raise important questions about how personal biases may influence professional advice and highlight the need for further training or guidelines on the appropriate use of convenience products in clinical practice.

Addressing the nutritional concerns regarding convenience cooking products is important, as non-users were more likely to view them as low in fibre and unbalanced, while users believed they provided adequate protein and could support weight control. The correlation between personal use and recommendation suggests potential bias, with frequent users perceiving these products more favourably, possibly downplaying concerns such as their low fibre and high sodium content. This may mean that recommendations are based on personal beliefs rather than objective assessment. Further research should examine these beliefs to ensure evidence-based dietary guidance is provided. Additionally, improving these products’ nutritional profiles, such as increasing fibre through including a variety of vegetables and reducing their sodium content, could enhance their acceptance among both health professionals and the public.

Another key aspect of this study was the examination of decision fatigue and its impact on food choices. Decision fatigue can occur when a person is confronted with the need to make an array of decisions throughout a day; this exhausts their ability to make further and sound decisions, which can lead to poorer decision-making [20,21]. Decision fatigue’s impact on food choices can mean that healthier decisions are less prioritised, and decision fatigue could lead to more impulsive and less health-conscious choices [35,36]. A substantial proportion of health professionals (86.7%) agreed that decision fatigue could lead to poorer food choices, and 61.5% believed that convenience cooking products could be a healthier option for individuals experiencing decision fatigue. This aligns with the theory that decision fatigue can impairs self-regulation and leads to less optimal food choices. When suffering from decision fatigue, it is easier for individuals to be influenced by advertisements, temptations, or convenience [37,38]. This is particularly apparent in food choice behaviours, where consumers are more likely to be influenced by a convenient food product or snack sitting at the check-out during shopping [39,40]. Given the significant cognitive load involved in daily food decision-making [41], convenience cooking products may offer a practical solution for time-poor individuals or those facing decision fatigue, potentially contributing to improved dietary quality in these populations if used appropriately. Further research is needed to explore the relationship between decision fatigue and the selection of convenience food options.

These findings align with broader discussions on the importance of food knowledge in clinical settings, as demonstrated in studies examining patient dietary literacy, such as the one by Petrelli et al. (2020), which highlights the role of nutritional education in shaping food choices, particularly in populations managing chronic conditions like type 2 diabetes [42]. From a clinical perspective, these findings suggest the need for more targeted educational interventions, not only for patients but also for health professionals. Research on dietary knowledge among individuals with a chronic disease highlights the gap between general nutrition awareness and the practical application of this knowledge in daily food choices. A similar concern arises regarding health professionals’ views of convenience cooking products: while some see them as a practical solution for time-poor individuals, others remain hesitant to recommend them due to concerns about their nutritional content. Therefore, integrating evidence-based guidelines on how to assess and select healthier convenience products could enhance the effectiveness of dietary counselling in clinical practice. Future research should explore structured strategies for incorporating convenience cooking products into dietary interventions while ensuring that their nutritional quality aligns with clinical recommendations.

This study highlights the complex relationship between health professionals’ personal behaviours, their professional recommendations, and the growing trend of using convenience cooking products. While these products offer practical benefits, there is a need for further education about their nutritional value and for improvements to make them healthier. Future research could explore how modifications to the ingredients and packaging of convenience cooking products could enhance their health benefits, potentially aiding in the promotion of vegetable consumption and better dietary practices in time-poor populations.

## 5. Conclusions

This study provides valuable insights into Australian health professionals’ perceptions and use of convenience cooking products. While the surveyed professionals acknowledge that these products offer practical benefits for busy lifestyles, some professionals reservations underscore the need for improved nutritional profiles to address concerns about their fibre, protein, and sodium content. The findings suggest that health professionals’ personal usage patterns influence their professional recommendations, emphasising the need for balanced, evidence-based guidance. As decision fatigue is recognised as a factor in food choice, convenience products may serve as a viable, balanced option for time-poor individuals if nutritional improvements are made. Future research should focus on reformulating convenience products to enhance their health benefits, potentially encouraging healthier eating habits among Australians with limited time or cooking skills.

### Limitations and Future Directions

This study provides novel insights into health professionals’ perceptions and use of convenience cooking products. However, several limitations should be acknowledged. First, the study utilised a convenience sample recruited through snowball sampling, which may limit the generalisability of the findings to the broader population of Australian health professionals. Future research should employ randomised or stratified sampling methods to enhance representativeness. Additionally, while this study highlights key differences between dietitians and nutritionists regarding their views on convenience cooking products, further qualitative research is needed to explore the underlying reasons for these differences. Understanding professional perspectives in more depth could help inform educational strategies or guidelines for recommending these products in clinical practice.

Future research should assess potential strategies to improve the nutritional profile of convenience cooking products, such as increasing fibre content, reducing sodium, and incorporating a greater variety of vegetables. Investigating the impact of reformulated products on both consumer acceptance and health outcomes could support evidence-based recommendations for their use.

## Figures and Tables

**Table 1 nutrients-17-01156-t001:** Demographics and profession (*n* = 143).

Confounder	Total	% of Subjects
**Gender**		
Male	5	3.5
Female	135	94.4
Others	3	2.1
**Age**		
18–24	15	10.5
25–34	76	53.1
35–44	32	22.4
45–54	12	8.4
55–64	6	4.2
**Qualification**		
Bachelor	55	38.4
Master	56	39.2
Other post-graduate qualification	23	16.1
PhD	9	6.3
**Profession**		
Dietitian (accredited practicing and other)	77	53.8
Nutritionist (associate or registered)	23	16.1
Nutritionist (other)	14	9.8
Nurse	15	10.5
Medical doctor	6	4.2
Other nutrition or allied health professional	8	5.6
**Engaging in profession**		
Yes	135	94.4
No	8	5.6
**Years of experience in profession**		
1–6	78	54.5
7–12	40	28.0
13–20	17	11.9
21–27	3	2.1
28–32	5	3.5

**Table 2 nutrients-17-01156-t002:** Health professionals’ use of convenience cooking products.

Do You Use These Products?	*n*	%
Yes	107	74.8
No	29	20.3
Maybe	7	4.9
**Frequency of using the products**		
Multiple times a week	23	16.1
Once a week	57	39.9
Once a month	23	16.1
Less than once a month	11	7.7
Never	29	20.3
**How often do you use these products**		
When I don’t feel like planning a meal	52	36.4
Just because	36	25.2
When I’m having a busy day	86	60.1
When I want to explore a new variety of food	43	20.1

**Table 3 nutrients-17-01156-t003:** Users and non-users of convenience cooking products among health professionals and recommendations to their patients/clients.

	Total		
Do you Recommend These Products		Users	Non-Users
	*n* (%)		
Yes	85 (59.4)	73 (64.1)	73 (41.4)
No	20 (14.0)	12 (10.5)	8 (27.6)
Maybe	35 (24.5)	26 (22.8)	9 (31.0)
Unsure	3 (2.1)	3 (2.6)	0 (0)
**How often do you recommend these products**		Users	Non-users
Multiple times a week	31 (21.7)	27 (26.7)	4 (19.1)
Once a week	27 (18.9)	22 (21.8)	5 (23.8)
Once a month	7 (4.9)	6 (5.9)	1 (4.8)
Less than once a month	5 (3.5)	2 (2.0)	3 (14.3)
Never	73 (51.0)	44 (43.6)	8 (38.0)
**Who Do you suggest these products to**		Users	Non-users
Families	75	69 (67.6)	6 (28.6)
Single households	66	46 (51.1)	13 (54.2)
Time poor people	112	95 (92.2)	18 (85.7)
People with little cooking skills	113	94 (92.1)	19 (90.5)
No one specific	13	12 (11.8)	1 (4.8)

**Table 4 nutrients-17-01156-t004:** Health perceptions regarding convenience cooking products.

Statement	Mean (95%CI)		*p*
	Non-Users	Users	
They provide adequate protein	2.2 (1.8–2.6)	2.6 (2.4–2.8)	0.05
They don’t have enough fibre	4.1 (3.8–4.5)	3.6 (3.5–3.8)	0.01
They can contribute to a balanced diet	3.1 (2.8–3.5)	2.6 (2.4–2.7)	0.0064
They can be useful for weight control	3.3 (2.9–3.6)	2.8 (2.6–3.0)	0.02

**Table 5 nutrients-17-01156-t005:** Decision fatigue statements.

Statement	Yes	No	Maybe	Unsure
	*n* (%)			
Do you think that a person who suffers from decision fatigue may be more prone to poorer food choices?	124 (86.7)	0 (0)	15 (10.5)	4 (2.8)
Do you think if a person suffers from decision fatigue the use of convenience cooking products could be a healthier more balanced option?	88 (61.5)	3 (2.1)	48 (33.6)	4 (2.8)
If you had a client/patient who was suffering from decision fatigue would you suggest these products to them?	77 (53.8)	5 (3.5)	50 (35.0)	11 (7.7)

## Data Availability

Data will be made available upon reasonable request, provided appropriate ethics clearances are obtained.

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
