# Peer review of "Health Professionals’ Knowledge and Views on the Use of Convenience Cooking Products: An Australian Cross-Sectional Study"

_nutrients, 2025, doi:10.3390/nu17071156_

Round 1

Reviewer 1 Report

Comments and Suggestions for Authors

This manuscript (nutrients-3541407) presents an interesting and relevant study investigating health professionals' perspectives on convenience cooking products. The study is timely, given the increasing reliance on such products due to lifestyle constraints. The methodology is sound, and the statistical analysis is well-performed. However, several areas require improvement, particularly in the clarity of the abstract, the consistency of terminology, the justification of some methodological choices, and the discussion of potential biases. Below are specific comments to enhance the manuscript.

  1. The abstract lacks clarity in certain sections, particularly in defining key terms and ensuring the results and conclusions are more structured. The sample size (n = 143) should be explicitly stated in the methods section of the abstract rather than within the results. The statistical significance of findings (p-values) should be reported in a structured format for clarity. For example: "Dietitians were less likely to recommend convenience cooking products (p = 0.0810) compared to nutritionists (p = 0.0750), suggesting professional differences in perception (χ2 = 44.48, P < 0.0001)."
  2. The study uses a snowball sampling method, which may introduce bias by limiting generalizability. However, this is not sufficiently discussed in the limitations section. The Materials and Methods section should include a brief justification explaining why this approach was used instead of a more representative sampling method. A statement in the Discussion should acknowledge the potential for sample bias and how it might impact the findings.
  3. Some reported p-values (e.g., p = 0.0810 and p = 0.0750) suggest trends but do not reach the conventional threshold for statistical significance (p < 0.05). However, they are discussed in a way that implies strong findings. Reframe these results using more cautious language, e.g., "While dietitians were less likely to recommend convenience cooking products than nutritionists, this association was not statistically significant (p = 0.0810)."
  4.  The study suggests that personal use of convenience products influences health professionals' likelihood of recommending them. However, it does not adequately explore whether this bias could impact patient outcomes. The claim that "86.7% of participants believed decision fatigue impacts food choices" is strong, but the mechanisms behind this association are not explored. Provide citations to support this claim or briefly explain how decision fatigue might contribute to poorer dietary choices.

Reviewer 2 Report

Comments and Suggestions for Authors

After some revisions, the manuscript submitted by Brasington et al. can be considered for publication in Nutrients.

The study’s aims and the applied methodologies need to be clarified in the abstract.

The Introduction section is well presented. However, the study’s novelty can be highlighted by the authors as well as the study’s objectives.

Regarding ethics, the study’s approval date is missing.

How did you establish your sample size and consider it adequate and representative of the study population? This must be clearly expressed in the manuscript.

The Results section is adequate but the Discussion needs to be improved. I suggest it is divided into subsections aligned with the Results. The study’s strengths and limitations must be analyzed, and more studies carried out in other regions of the world should also be cited and discussed properly.

The Conclusions are fine.

Reviewer 3 Report

Comments and Suggestions for Authors

Dear Authors,

we hope you'll appreciate the suggestions.

Best

Round 2

Reviewer 1 Report

Comments and Suggestions for Authors

The authors addressed all my previous concerns. The current manuscript is improved compared to the initial submitted version.

Reviewer 3 Report

Comments and Suggestions for Authors

Dear Authors,

very good job.

In this form ready for publication.

Best